# Are Users Good Assessors of Social Dominance in Domestic Horses?

**DOI:** 10.3390/ani14131999

**Published:** 2024-07-07

**Authors:** Ewa Jastrzębska, Marta Siemieniuch, Adriana Bizio, Julia Pietruszka, Aleksandra Górecka-Bruzda

**Affiliations:** 1Department of Horse Breeding and Riding, Faculty of Animal Bioengineering, University of Warmia and Mazury, Oczapowskiego 5, 10-719 Olsztyn, Poland; e.jastrzebska@uwm.edu.pl (E.J.); adrianna.bizio@student.uwm.edu.pl (A.B.); julia.pietruszka.1@student.uwm.edu.pl (J.P.); 2Research Station of the Institute of Animal Reproduction and Food Research of PAS, Popielno 25, 12-220 Ruciane-Nida, Poland; 3Department of Animal Behaviour and Welfare, Institute of Genetics and Animal Biotechnology, Polish Academy of Sciences, 05-552 Jastrzębiec, Poland

**Keywords:** horses, social dominance, agonistic/aggressive behaviours, submissive behaviours, riding instructors, equine professionals

## Abstract

**Simple Summary:**

It is vitally important that horse riding instructors know which horses are dominant and which are submissive to prevent injuries resulting from social conflict during rides. Eight equine practitioners were asked to rank 20 horses according to their behaviour known from previous work and non-work observations. In pairs, the horses were also confronted with a limited amount of hay to test their dominance and associated behaviours. The time spent eating hay and submissive behaviours in the test were strongly related to each other, showing that dominant horses were eating hay for a longer time than submissive ones. The raters were moderately concordant in their ranking of horses, but when familiar with a group of horses they were able, consciously or unconsciously, to detect the dominance–submissiveness relationships between animals. This is key to assuring the safety of both horses and riders.

**Abstract:**

Horse users and caretakers must be aware of the risks of mixing social groups. The current study investigated whether eight equine practitioners can assess the social dominance rank of 20 horses. The horses’ feeding time and agonistic/aggressive and submissive behaviours were observed during the feed confrontation test, and the dominance index (DI) was calculated. Kendal’s W, Spearman correlations and factor analysis were applied to test the raters’ agreement, the relationship between dominance ranks and the behavioural variables, and to determine the clustered behaviours. The agreement between all raters in the classification of dominance order ranged from moderate to perfect. The ranking by every rater was strongly and negatively correlated with the time of eating in feed confrontation tests and with the DI, evidencing shorter feeding times for more submissive horses. The withdrawal of the horse when threatened was the behavioural variable that was most often correlated with raters’ ranking. The current study confirmed the abilities of practitioners to categorise the horses under their care according to their social interactions. Additionally, rolling when denied access to feed was proposed as frustration-releasing (redirected) behaviour.

## 1. Introduction

For many years, horses, like other farm animals, were treated objectively so that people could benefit from their work without taking into account the horses’ comfort in life. The rearing, nutrition, and prevalent activities involving horses are greatly influenced by the availability and productivity of grasslands and farmlands. Horses are increasingly being classified as companion animals [1,2], and their welfare as farm animals is relevant and deemed scientific consideration. The change in perception of equine welfare by owners, trainers, riders, and grooms has also been observed [3]. However, it is advisable that people who own and work with horses have a minimum knowledge and abilities to observe them to assess and interpret their behaviour appropriately. 

Horses, as social animals, live in groups. In natural conditions or in the wild, they form family groups (harems) consisting of several adult females (mares), a male (stallion), and their young offspring, who leave their familial group at the age of about two years [4]. Sometimes, the herd may include a subordinate adult stallion; however, males without mares usually form bachelor herds. Both in the case of harems and bachelor herds, horses create a social structure based on dominance and submission [4]. When there are many horses, harems can aggregate in large herds [5]. Such herds move together, as observed in other equids (zebras and Przewalski’s horses), while maintaining the distinction between harems and bachelor herds, as demonstrated by Ozogany et al. [6]. Therefore, horses are capable of learning and following complex social rules at the social group and supergroup level.

In social groups, horses form and maintain relationships based on affiliative behaviour as well as the dominance–submissiveness system. While some researchers claim that the social hierarchy in horses is linear [4], others promote a non-linear system [7]. Horses can achieve or change their social position depending on age or the order of arrival to the established group, gender, group composition [4], dam’s social rank [8], the presence of supporting pair mates [9], and horse’s predisposition to dominance and submissiveness [7]. Social conflicts in horses may result in injuries [10,11]. The horses fight for attractive resources that are available in limited quantities, such as tasty feed. They defend access by biting or kicking, which effectively drives away subordinate individuals. Notwithstanding the latter, the horses tolerate chosen individuals approaching them when feeding or beyond feeding times, while other individuals are not welcomed and thus threatened or actively chased away [7]. Knowledge of mutual relationships in the group also allows horses to effectively avoid threats by maintaining appropriate individual distances from given individuals. 

When space is limited, such as when horses are kept in boxes, small paddocks, or used for riding, horses often cannot withdraw on a warning or the aggressive behaviour of conspecifics. Therefore, horse users and caretakers must be aware of the risks that may arise when mixing social groups in housing, pasturing, or riding conditions. Horses for group riding are often randomly matched based on the rider’s skills rather than the horses’ social relationships. This is essential when riding in a single file, as the horse should not attack the one preceding it. In fact, horse users or caretakers may not know how to assess the social dominance that exists within a group of horses. However, to a large extent, experienced instructors and horse owners can ensure the safety of horses and riders. 

This study aimed to check whether equine practitioners, observing horses during handling and riding, can assess their social dominance rank and how their assessment is related to the behaviour of horses when contesting an important resource. Validation of practitioners’ ratings of the behaviour of horses would confirm their ability to predict dangerous situations during the preparation of horses for the ride and the ride itself. 

## 2. Materials and Methods

### 2.1. Animals

A total of 20 riding horses (5 mares and 15 geldings, aged from 3 to 22 years) belonging to the riding school of the University of Warmia and Mazury in Olsztyn, Poland, were used for the present study. In this riding school, the horses serve for riding for 15.0 ± 1.69 years. The horses were in good body condition, and no clinical signs of lameness or other diseases were observed. The horses were mainly used for leisure riding in the arena (for 2 h daily, 5 days per week). They were maintained in straw-bedded boxes (9 and 12 m^2^), fed with hay and oat thrice daily, and were turned out for a sandy paddock (in autumn and winter) or a pasture (in spring and summer) for 5 h a day. For at least six months, the studied horses were turned out together in one group, so they were familiar with each other. 

### 2.2. Feed Confrontation Test

The tests were conducted between 11 a.m. and 2 p.m., for 10 consecutive days, with a break for the weekend. In pairs (each horse with each), the horses were exposed for 5 min to a small amount of hay (50 × 50× 50 cm) on the ground. The hay was used to avoid excessive aggression when competing over access to more tasty feed, but placing it on the ground was considered competitive enough [12]. The tests were conducted in a familiar covered riding arena of 18 × 50 metres (Figure 1). The horses in pairs, selected randomly from the tested population, were released in the arena. After 5 min, the horses were taken back to the stable while another pair was released. The experimenter, located in the corner of the arena at a distance of 10 m, registered the behaviour of horses with the video camera during each of the 190 confrontations. Thereafter, the behaviour of horses was categorised according to Table 1, and the duration of feeding and occurrences of behavioural variables were measured from the recordings. In addition, the horses were classified as “longer time” (L) consumers and “shorter time” (S) consumers, according to which horse in the pair was eating for a longer time during each confrontation. “Same time” (SS) consumers were the horses that fed for an equal time, denied (D) horses were not allowed to eat by the other horse in the pair, and “non-interested”(NI) horses were the horses not interested in eating at all but stood still or travelled around the arena. The dominance index (DI), adapted from Lehmann et al. [13], was calculated for each horse as:DI = (L + SS) − (S + D)/L + SS + S + D
This calculation gives a value ranging from −1 to 1. Considering that it was planned to detect horses sharing the food, SS consumers were added to L as they were equally successful in getting access to feed.

### 2.3. Classification of Horses According to Dominance

Based on their daily experience, four instructors, three undergraduate students and one caretaker (employees and students of Warmia and Mazury University in Olsztyn, Poland) familiar with horses (raters, anonymised as A–H) were asked to rank the horses (without any consultation) from the most (position 1) to least (position 20) dominant. All but one rater had basic ethological knowledge (teachers or undergraduate students of the Faculty of Animal Bioengineering), but they were not familiar with the methods of dominance assessment for horses. 

Four raters participated in the confrontation test (“test” raters, A–D, one teacher and three students, all females) involved in the fulfilment of a Master’s thesis. The other four (three teachers and one caretaker) were ignorant of the behaviour of horses in the test (“ignorant” raters, E–H, two females and two males). All of them were asked to complete a simple survey (Appendix A) on how they deal with horses daily to ensure their safety during handling and riding (for riders’ safety questions, only instructors filled out the form).

### 2.4. Statistical Analyses

The agreement between raters in ranking horses according to their dominance was tested with Kendall’s W test for “test” raters, for “ignorant” raters and for all raters. Due to the non-normal distribution of the behavioural variables, non-parametric statistics were applied. The duration of feeding and the occurrence of behaviours were summed for each horse-horse confrontation in the feed test, giving the totals of the behaviours for each horse in all 190 tests. Spearman correlations of DI, rater’s ranking, and behavioural variables were used to check whether they predict the rater’s ranking based either on previous experience with horses only or, additionally, on participation in feed tests. Lastly, factor analysis with varimax rotation was used to identify the groups of behaviours presented in the confrontation test. The SAS statistical package (SAS Institute Inc., Cary, NC, USA) was used for all analyses except for Kendall’s *W*, which was calculated in Microsoft Excel v. 13.

## 3. Results

### 3.1. Raters’ Agreement in Dominance Rank 

The agreement between all raters in the classification of dominance order was moderate (Kendall’s *W* = 0.65, *X*^2^ = 98.1, *p* < 0.01); however, for “test” raters, it was almost perfect (Kendall’s *W* = 0.97, *X*^2^ = 73.6, *p* < 0.01) and higher than for “ignorant” raters (Kendall’s *W* = 0.57, *X*^2^ = 43.4, *p* < 0.01).

### 3.2. Raters Dominance Rank, Dominance Index and Behaviour in Feed Confrontation Test

The descriptive statistics for the occurrence and durations of the behaviours (totals) observed in the feed confrontation test are presented in Table 2.

In a total of 190 confrontations, one of the horses was not allowed to share the feed with the dominant horse in only 24 tests (Figure 2).

The correlations between the raters’ domination rank, DI and the behavioural variables are shown in Table 3. The ranking by every rater was strongly and negatively correlated with the time of eating in feed confrontation tests and, consequently, with the DI (except for a tendency for rater F), evidencing shorter feeding times for more submissive horses (from *r_s_* = −0.48, *p* = 0.03 to *r_s_* = −0.78, *p* < 0.01 for eating and from *r_s_* = −0.41, *p* = 0.07 to *r_s_* = −0.65, *p* < 0.01 for DI).

Unexpectedly, it was not aggressive behaviour (Figure 3) but submissive behaviour, i.e., withdrawal when threatened (from *r_s_* = 0.66 to *r_s_* = 0.82, *p* < 0.01), that was the behavioural variable most often correlated with the raters’ ranking. The occurrence of sniffing was also predictive of the horse’s rank, either significantly (raters A, B, C, F, and H, from *r_s_* = 0.45, *p* < 0.01 to *r_s_* = 0.54, *p* = 0.05) or at the tendency level (raters D, E, and G, from *r_s_* = 0.38, *p* = 0.09 to *r_s_* = 0.43, *p* = 0.06). The horses that allogroomed during the test were ranked lower (rater H: *r_s_* = 0.45, *p* = 0.05, rater B: *r_s_* = 0.42, *p* = 0.07 and rater C: *r*_s_ = 0.39, *p* = 0.09, the latter two on the tendency level). According to the ranking of rater E, the horses that were observed rolling had a lower dominance rank (*r_s_* = 0.50, *p* = 0.03), and the ranking of rater C tended to correlate with the number of snorts (*r*_s_ = 0.39, *p* = 0.09). The horses ranked as less dominant by the rater F and H (*r*_s_ = 0.57, *p* < 0.01 and *r_s_* = 0.54, *p* < 0.01, respectively) as well as B and D (at tendency level, *r_s_* = 0.42, *p* = 0.06 and *r_s_* = 0.44, *p* = 0.06, respectively) kicked off other horses and squealed more (*r_s_* = 0.44, *p* = 0.05 and *r_s_* = 0.47, *p* = 0.03, for raters F and H, respectively) during the test.

### 3.3. Factor Analysis of Behavioural Variables during Feed Confrontation Test

The first four factors with an eigenvalue of more than 1 explained 78.5% of variance (Table 4). The variables with loadings above 0.50 were retained. The F1 group grouped aggressive behaviours such as kicking, backing, squealing, and moving the ears back (loadings from 0.81 to 0.91). The F2 was loaded with submissive or comfort behaviours like retreat, rolling, grooming, and sniffing (loadings from 0.61 to 0.90), as well as (negatively) with eating time (−0.91). Driving and biting are loaded on factor 3 (both loadings are 0.86), while the “wet dog” shakes and snorts are on F4 (0.89 and 0.88, respectively).

### 3.4. Raters’ Survey

The instructors’ experience with horses ranged between 20 and 30 years (mean = 24.2, *s.d.* = 4.35). They declared that they spent from 10 to 30 min (mean = 21.0, *s.d.* = 6.41) on observation of the horses beyond the riding lessons (Appendix A). Two of the instructors declared that they take into consideration both the riders’ skill and the behaviour of horses when matching the particular horse–rider pairs, while one indicated that the rider’s skills are most important in such matching. However, all instructors agreed that they always decide upon the order of horses in a row in group riding. All raters declared that it is easy for them to identify both the aggressive and submissive horses, as they have experience and sufficient knowledge of equine behaviour. They learned about it both from scientific sources (academic education) and from equine “behaviourists” (the trainers of “natural horsemanship” or similar). Interestingly, some of the “test” raters declared that the observation of horses during feeding confrontation was novel and important knowledge despite their long-time experience with these horses. 

## 4. Discussion

The present study showed that horse users are able to rank the horses in an order that reflects their social behaviour. This rank was mirrored in the dominance index and the time of eating hay (as the two variables were interrelated), even if, in the majority of cases, the horses shared the feed. Interestingly, and contrary to expectations, the submissive behaviour was strongly and incontestably related to the dominance rank of all raters, while the dominant and aggressive behaviour was definitively less. Experienced “horse” people can consciously and/or unconsciously understand subtle aspects of horse behaviour and social relationships within the group.

Identifying and interpreting the behaviour of another species is an important skill in the animal world. Animals distinguish complex social relationships and interpret the body signals and gestures of other species [18]. This ability is also typical of humans [16]. When using animals, people observe and interpret their behaviour and its consequences [19]. Although they may be biased by anthropomorphic ways of interpreting animal behaviour [20,21], the observations of equine practitioners serve for pain detection [22], personality assessment [23,24,25], trainability [26], rideability [27], welfare [28], and other goals. Caretakers and trainers can assess the different aspects of their animals’ behaviour since they accumulate knowledge about animals based on everyday contact [19]. In the present study, all raters declared that they are interested in the behaviour of horses not only when working with them but also beyond working hours. This allows them to easily identify the individuals that may be difficult for beginning riders. Our raters agreed in their assessment of the horses’ dominance, although their assessment, “predicting” mostly similar behaviours, differed slightly. Unfortunately, it is difficult to say based on our simple survey why there were slight differences between the raters, especially since everyone answered the questions similarly. Presumably, these are differences in the ability to analyse observed behaviour, or the linear hierarchy or the pair test does not fully correspond to the social hierarchy in the entire group of horses. 

There is, however, a risk that the owners, handlers or caretakers can miss important information, which is the case of pain assessment. The horses studied in equine hospitals showed more pain-related behaviours in the absence of humans [29]. It may be that people focus on a specific behaviour and miss others which are equally important. The current study showed that the practitioners engaged in testing were exceptionally consistent in their rankings. Moreover, “ignorant” raters’ rankings (especially the raters F and H) were correlated to more behaviours occurring during testing about which the ignorant raters were not aware. The generalisation by humans of complicated social relationships within this group, as well as the knowledge gathered from scientific and practical sources, had applied importance in the management of riding lessons with riders of different skills. 

Despite the horses mostly sharing the hay, the current results confirmed that the eating time, as well as the DI, was a good indicator of the dominance within the pair. Although it cannot be translated to social relationships at the group level, it was highly predictive of raters’ dominance ranking for all horses. The horses ranked lower were evidently more yielding in the test, and this behaviour was the best predictor of their position in rankings. It was rather surprising, as the aggressive behaviour is usually more evident to observers, but in the present study, the latter were only moderately correlated to rankings. Since the horses were familiar with each other for a long time, the very subtle warnings, or maybe the knowledge about the position of the opponent, were sufficient for the horse to withdraw or not approach when the dominant was feeding after unsuccessful trials. As proposed by [7], an “avoidance” or “submission” order reflects better the social system in stable equine groups. 

The aggressive and submissive behaviour clustered separately confirmed the potential of the confrontation test, although not perfect at the group level, which was helpful in the quick revealing of dominant relationships in horses. It is striking that the biting and chasing of other horses clustered separately and not with other agonistic behaviour. These behaviours can be characterised by the horse’s part of the body, which is engaged in conflict, and the biting and chasing logically occurring together [14]. The same could be said about the kicking and backing, which could explain the loadings in terms of separate factors.

The shorter time of feeding was strongly associated with submissive behaviour, as well as sniffing, grooming, and rolling. The sniffing was observed in horses that were not so much interested in eating as in interacting with each other. The same was observed for mutual grooming, as it occurred only when the horses were not interested in feeding, and they were both affiliatively disposed. These were horses that showed an evident willingness to interact, and the fact that they ate for less time was not due to aggressive behaviour. The main reason for the shortened eating time was the preference for interacting with overeating. In this context, the usual feed, which is hay, may not have been as attractive as other tasty feed [30], and thus possibly did not clearly indicate social dominance. Instead, the hay was a good feed to observe sharing it between the horses, although sharing the hay lying on the ground provoked the most frequent agonistic behaviour in an earlier study [12]. Sharing grains was observed in Przewalski’s horses, but it was most often for sires or mares sharing it with their foals (36–77% for dams and sires, respectively) as compared to other adult horses (7%) [31]. Some horses in the current study were not interested in feeding due to separation stress. Despite not being completely socially separated, one gelding was observed to be stressed in the “incomplete” social group it usually turned out with. This confirms the previous study, when the heart rate of a separated horse decreased proportionally to the number of horses left for the company [32], confirming the ability of horses to generalise on the group “completeness”.

Contrarily, rolling was mainly shown by horses denied access to feed, which can suggest it can be a potential indicator of frustration. So far, this behaviour has been interpreted as comfort behaviour [7,33]. Actually, the “wet dog” shake almost always follows rolling behaviour, and in the present study, the “wet dog” shake and sneezing/snorting were loaded together on the last factor. Since sneezing/snorting is classified as a relaxed behaviour [34], these two behaviours indeed may have been exhibited in relaxed horses. However, the studies [35,36] showed that horses were more willing to roll after the increased workload, while no effect of exercise was found in [37]. The current study is the first to show that rolling was exhibited by the horses driven off the feed. It can be hypothesised that rolling, like yawning [38], except for its comforting function, may be shown by horses after physical and/or mental discomfort (displacement behaviour). This hypothesis may be partly supported by the results of [39], who found that rolling in straw provided as an environmental enrichment was observed only in stereotypic horses as compared to non-stereotypic ones. Further studies are required to confirm the hypothesis.

The practical outcomes of the study still need various questions to be answered in further studies. While instructors/riders may be aware of dominance ranks in their group of horses, during daily tasks, there may be possible confounding factors, such as the number of horses during each class, as usually there are no pairs, but multiple horses in the same arena. Also, the presence of people can influence horse behaviour, like the instructors who decide about the horses in line or tasks to do. Moreover, the horses should be trained to work with other horses. Based on their experience, the instructors have to warn the naïve or begging riders about a potential risk of conflict between given horses. Also, it is interesting how the experienced rider’s high control of the horse’s behaviour may influence the possible dominance-related signals or behaviour during the ride. 

The main limitation of this study is using a pair test, as its results may not represent the complicated structure of the equine group. Despite the feed competition test being criticised for producing dominance rank results that do not reflect the true relationships in the whole group [9,40,41] or the specific conditions of working with horses in the riding centre involving a rider-horse adjusted selection of the horses for the group rides, it was chosen because it provides information closest to riding centre conditions. Moreover, since the dominance relationship was defined as “the pattern of repeated, agonistic interactions between two individuals characterised by the number and default yielding response of the opponent rather than escalation” [42], and was used in studies on horses [8,9,13,41], it was decided to apply it in the present study. Also, the linear order of social dominance may not represent a true social net in horses. However, it was the simplest method of assessing dominance for raters.

## 5. Conclusions

The current study confirmed the abilities of practitioners to categorise the horses under their care according to their social interactions. The practitioners with appropriate education and practice were consistent in their ratings, which makes them appropriately prepared for their work in teaching equitation, assuring the safety of both riders and horses. Given the progressing automatization of farm animal behaviour assessment [43], the attentive observation of the animal by its caretaker cannot be overvalued. However, in any case of human assessment of animal behaviour, we should always be aware that we can miss a range of signals that are not available to human senses, and classifications of any type may bear our human-skewed interpretations.

An interesting outcome of the present study is the rolling observed in horses denied feeding by their more dominant counterparts. This behaviour in such a situation may be proposed as frustration-releasing (redirected) behaviour.

## Figures and Tables

**Figure 1 animals-14-01999-f001:**
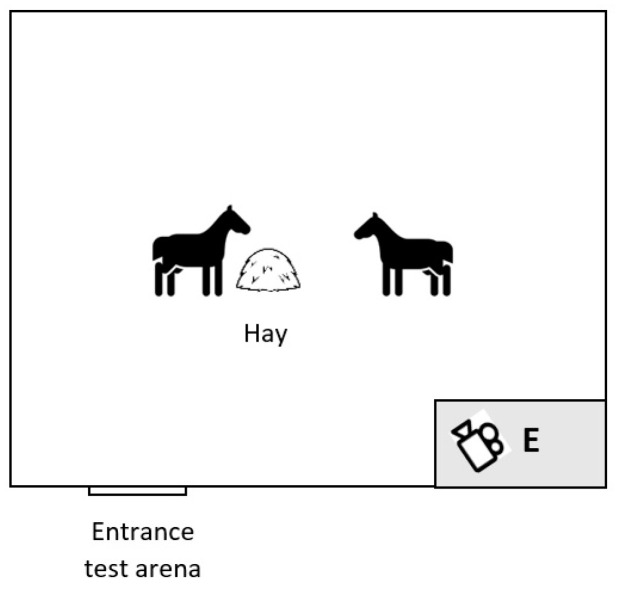
Schematic representation of the test area. E: experimenter recording the behaviour of horses.

**Figure 2 animals-14-01999-f002:**
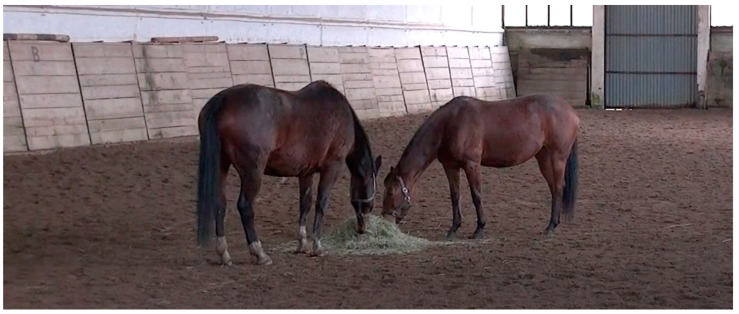
The study horses sharing the hay in a feed confrontation test.

**Figure 3 animals-14-01999-f003:**
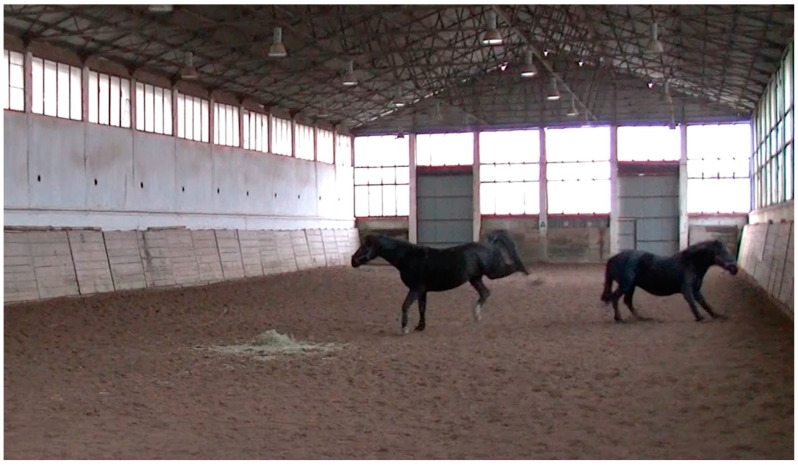
Aggressive behaviour of studies horses during the feed confrontation test.

**Table 1 animals-14-01999-t001:** Definitions of behavioural variables during food confrontation test and the measurement method.

Behaviour	Definition
Eating time	The time from grasping the hay with the lips to stopping the chewing.
Ears (pinned) back	The horse moves and holds the ear pinnae backwards without changing the position of the body.
Biting/biting threats	One horse, with its ears pinned back, bodily contacts another horse by retracting its lips and closing its teeth on the other horse’s body. If the hold is sustained, this behaviour may be classified as a ‘grasping’ behaviour. One horse performs biting-like movements towards another horse without making physical contact. This behaviour includes directed head movements, with the neck extended and ears pinned back [14].
Squealing	High-pitched vocalisation of variable loudness and typically of less than 1 s. The head can be in a variety of positions, and the mouth is typically closed during the squeal. These vocalisations are typical during olfactory investigation, posturing, biting, and nipping, as well as both mock and serious fighting [15,16].
Kicking	Rapid hindleg extension with contact. With its ears pinned back, one horse rapidly extends one or both hind legs backwards toward another, resulting in physical contact between the aggressor’s hooves and the other horse’s body [14].
Chasing	With its ears oriented backwards, one horse initiates a fast-paced pursuit of another horse, spanning at least three strides [14].
Backing	A horse engages in retrogressive locomotion, moving backwards towards another horse with its ears pinned back [14].
Retreat	One horse moves to maintain or increase the distance from a threatening approaching horse, either at a walk or trot. Both horses orient their ears backwards. A retreat can be differentiated from flight by the slower speed [14].
Sniffing	One horse sniffs various parts of another horse’s body, such as the head, neck, flank, genitals, tail, or perineal region. The second horse may reciprocate this behaviour. The valence of this interaction varies depending on the context and can be discerned through accompanying vocalisations, stomping, or ear position [14]. In the current observations, this behaviour had an affiliative valence.
Grooming	Two horses standing close, head-to-tail or head-to-head, with their ears oriented forward or laterally, employ their teeth, lips, or tongue to clean and maintain each other’s bodies [14].
Rolling	The horse lies down on one side and then rotates its trunk with a lateral pushoff to attain a supine position. It holds this fixed position while it performs one or two slight rubs off its back on the chosen substrate. By laterally swinging its legs, it then falls back on its original side [7].
“Wet dog” shake	After rolling, the horse rises, stands squarely, and then performs the ‘wet dog shake’. In this, a wave of rippling skin passes from the head to the hindquarters, the legs and tail, dislodging matter from the hide and hair. The vigour in the shake is such that the limbs quiver down to the hooves [7].
(Sneezing) snorting	Clearing the nostrils of phlegm, flies, etc. [17].

**Table 2 animals-14-01999-t002:** Descriptive statistics of behavioural variables (^1^ duration in minutes, ^2^ numbers per five minutes) during the feed confrontation test (totalled for all tests).

Behaviour	Mean	s.d.	Median	Q1; Q3	Range
Eating ^1^	58.0	17.9	60.8	49.4; 72.2	8.94–79.6
Ears back ^2^	26.1	20.9	21.0	8.50; 42.0	1.00–71.0
Biting ^2^	12.6	18.5	6.00	1.50; 15.0	0.00–71.0
Squealing ^2^	8.05	9.14	6.00	2.00; 10.0	0.00–37.0
Kicking ^2^	9.25	9.49	7.00	2.50; 11.5	0.00–33.0
Chasing ^2^	3.00	8.05	0.00	0.00; 2.50	0.00–36.0
Backing ^2^	14.0	13.4	10.5	4.00; 20.5	0.00–54.0
Retreat ^2^	18.7	15.7	17.0	3.50; 33.5	0.00–47.0
Sniffing ^2^	36.1	15.3	35.0	24.0; 52.0	15.0–69.0
Grooming ^2^	3.30	5.41	1.00	0.00; 5.50	0.00–22.0
Rolling ^2^	3.10	3.80	2.00	0.00; 4.50	0.00–14.0
“Wet dog” shake ^2^	0.25	0.75	0.00	0.00; 0.00	0.00–3.00
(Sneezing) snorting ^2^	0.95	1.57	0.00	0.00; 1.50	0.00–5.00

**Table 3 animals-14-01999-t003:** Spearman correlations (r_s_, *p*) between raters’ dominance rank, dominance index (DI) and the behaviour of horses during the feed confrontation test. Significant (*p* ≤ 0.05) correlations are in bold, and correlations at tendency level (*p* < 0.10) are in bold and italics.

Rater Behaviour	A	B	C	D	E	F	G	H
DI	**−0.61** **(<0.01)**	**−0.63** **(<0.01)**	**−0.63** **(<0.01)**	**−0.60** **(<0.01)**	**−0.65** **(<0.01)**	* **−0.41** * * **(0.07)** *	**−0.55** **(0.01)**	**−0.57** **(<0.01)**
Eating	**−0.68** **(<0.01)**	**−0.67** **(<0.01)**	**−0.69** **(<0.01)**	**−0.69** **(<0.01)**	**−0.78** **(<0.01)**	**−0.58** **(<0.01)**	**−0.48** **(0.03)**	**−0.64** **(<0.01)**
Ears back	0.14 (0.54)	0.09 (0.69)	0.08 (0.72)	0.19 (0.41)	0.09 (0.69)	0.20 (0.41)	0.11 (0.64)	0.11 (0.63)
Biting	−0.10 (0.67)	−0.11 (0.64)	−0.07 (0.77)	−0.21 (0.38)	−0.19 (0.42)	−0.12 (0.62)	−0.12 (0.60)	0.16 (0.49)
Squealing	0.20 (0.39)	0.31 (0.18)	0.18 (0.45)	0.02 (0.21)	0.26 (0.26)	**0.44** **(0.05)**	0.26 (0.28)	**0.47** **(0.03)**
Kicking	0.34 (0.14)	* **0.42** * * **(0.06)** *	0.29 (0.22)	* **0.44** * * **(0.06)** *	0.35 (0.12)	**0.57** **(<0.01)**	0.39 (0.09)	**0.54** **(0.04)**
Chasing	0.03 (0.89)	−0.03 (0.88)	−0.02 (0.94)	−0.04 (0.94)	0.36 (0.12)	−0.29 (0.21)	−0.15 (0.54)	−0.26 (0.27)
Backing	0.24 (0.30)	0.30 (0.20)	0.18 (0.44)	0.33 (0.15)	0.20 (0.39)	* **0.42** * ** *(0.06)* **	0.17 (0.46)	0.34 (0.14)
Retreat	**0.68** **(<0.01)**	**0.66** **(<0.01)**	**0.71** **(<0.01)**	**0.71** **(<0.01)**	**0.82** **(<0.01)**	**0.75** **(<0.01)**	**0.67** **(<0.01)**	**0.76** **(<0.01)**
Sniffing	**0.46** **(0.04)**	**0.53** **(0.02)**	**0.54** **(<0.01)**	* **0.43** * * **(0.06)** *	* **0.38** * * **(0.09)** *	**0.49** **(0.03)**	* **0.42** * ** *(0.06)* **	**0.45** **(0.05)**
Grooming	0.30 (0.19)	* **0.42** * ** *(0.07)* **	* **0.39** * * **(0.09)** *	0.30 (0.19)	0.37 (0.10)	0.36 (0.11)	0.26 (0.26)	**0.45** **(0.05)**
Rolling	0.22 (0.34)	0.20 (0.38)	0.30 (0.20)	0.27 (0.24)	**0.50** **(0.03)**	0.26 (0.27)	0.18 (0.44)	0.28 (0.22)
“Wet dog” shake	0.15 (0.52)	0.20 (0.40)	0.17 (0.46)	0.22 (0.33)	0.29 (0.22)	0.25 (0.29)	0.25 (0.29)	0.27 (0.24)
(Sneezing) snorting	0.36 (0.12)	0.31 (0.18)	* **0.39** * ** *(0.09)* **	0.34 (0.14)	0.17 (0.48)	0.20 (0.41)	0.28 (0.24)	0.25 (0.28)

**Table 4 animals-14-01999-t004:** Factor analysis of behavioural variables during the food confrontation test. Bolded are loadings above 0.50.

Factor Loadings Behavioural Variable	F1	F2	F3	F4
Kicking	**0.91**	0.06	−0.16	−0.02
Backing	**0.90**	0.04	−0.18	−0.14
Squealing	**0.90**	0.21	−0.03	0.10
Ears back	**0.81**	−0.07	−0.21	−0.04
Retreat	0.12	**0.90**	−0.15	0.08
Rolling	−0.15	**0.71**	−0.33	0.09
Grooming	0.29	**0.65**	0.14	0.14
Sniffing	−0.27	**0.61**	0.42	−0.10
Eating	−0.09	**−0.91**	−0.18	−0.07
Chasing	−0.13	−0.6	**0.86**	0.11
Biting	−0.35	0.20	**0.86**	−0.07
“Wet dog” shake	−0.05	0.13	−0.24	**0.89**
(Sneezing) snorting	−0.03	0.10	0.30	**0.88**
% variance explained	33.4	20.0	13.0	12.1

## Data Availability

The data presented in this study are available on request from the corresponding author.

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
