# Peer review of "Are Users Good Assessors of Social Dominance in Domestic Horses?"

_animals, 2024, doi:10.3390/ani14131999_

Round 1

Reviewer 1 Report

Comments and Suggestions for Authors

Dear Authors,

thank you for submitting your manuscript “Are users good assessors of social dominance in domestic horses?”, presenting the results of your project aimed at evaluating the ability of equine practicians in assessing social dominance in horses.

The paper addresses a undoubtedly relevant topic for the equine sector and may pose the basis for further research on the argument.

While I found your manuscript of interest and well written, I have some minor comments, specified below.

Introduction

LL45-48 These sentences may lead to the interpretation of minor importance of farm animal welfare and that horse welfare is relevant only because they are more frequently classified as companion animals rather than production animals. While welfare of companion animals is certainly important for owners, all animals’ welfare is relevant and deem scientific consideration.

LL70-71 While I understand the choice of the verb “eliminate”, I believe it could be misleading. Subordinate horses are not meant to be physically eliminated. Physical confrontations have, indeed, the aim of creating a ranking that, being respected by all subjects, remove the need for fights. In a stable group, fights are reduced if not eliminated.

LL76-85 This paragraph would require references.

Material and Methods

LL98-105 Some more information are needed: did each horse known all the other subjects? Did they encountered prior to the test? How were groups constructed during turns out? How familiar was each horse with the others?

LL120-124 You paired each horse with each. You have a total of 20 horses and the order of pairing (A vs B or B vs A) is not relevant. Therefore, you have a combinations issue (the way of selecting objects or numbers from a group of objects or collections, in such a way that the order of the objects does not matter – how many couples can we create having N horses?). To calculate it, the formula is nCr = n!/[r! (n-r)!], with nCr representing the number of combinations from “n” objects taken “r” at a time. So, in your case, 20!/2!(20-2)!=190 confrontations. You state you did 380 confrontations. What am I understanding incorrectly?

LL148-149 How did you assess that the professionals had “basic ethological knowledge”?

LL152-154 The questions included in the survey should be available for the reader.

L162 You state that you performed 379 tests, but the number was 380 in line 124.

Results

L184 You state that you performed 379 tests, but the number was 380 in line 124.

Table 3 P values are reported both as precise number and <0.01. I would choose one format only. In the caption, you reported that correlation at tendency level is “P<10”. Do you mean P<0.1?

LL242-243 What “scientific sources” means in this context? Scientific papers? Textbooks? Others? I wouldn’t call “the trainers of ‘natural horsemanship’ or similar” behaviourists. They are trainers and their scientific competencies are diverse and often debatable.

Discussion

I would like to see some discussion regarding practical applications of your findings. While instructors/riders may be aware of dominance ranks in their group of horses, during daily tasks there may be possible confounding factors, such as: number of horses during each class (there are no usually pairs, but multiple horses in the same arena), presence of people (do people influence horse behaviour? In which way? Do they influence their dominance/submission behaviour?), training (are horses trained to work with other horses, no matter the dominance ranks? Since the rider requests drive the horse behaviour, how does this influence the possible dominance related signals or behaviour?).

References

Please check the references section for homogeneity.

Author Response

Reviewer 1

Dear Authors,

thank you for submitting your manuscript “Are users good assessors of social dominance in domestic horses?”, presenting the results of your project aimed at evaluating the ability of equine practicians in assessing social dominance in horses.

The paper addresses a undoubtedly relevant topic for the equine sector and may pose the basis for further research on the argument.

While I found your manuscript of interest and well written, I have some minor comments, specified below.

 Thank you very much for your positive opinion. We hope the manuscript was improved.

Introduction

LL45-48 These sentences may lead to the interpretation of minor importance of farm animal welfare and that horse welfare is relevant only because they are more frequently classified as companion animals rather than production animals. While welfare of companion animals is certainly important for owners, all animals’ welfare is relevant and deem scientific consideration.

These lines were corrected as recommended.

LL70-71 While I understand the choice of the verb “eliminate”, I believe it could be misleading. Subordinate horses are not meant to be physically eliminated. Physical confrontations have, indeed, the aim of creating a ranking that, being respected by all subjects, remove the need for fights. In a stable group, fights are reduced if not eliminated.

We agree, the word ‘eliminate’ doesn’t fit here. It has been changed to “drive away”.

LL76-85 This paragraph would require references.

As these are practical observations, we think these lines do not need referencing. Even though instructors may not have in-depth knowledge of social interaction between horses, they are often good observers.

Material and Methods

LL98-105 Some more information are needed: did each horse known all the other subjects? Did they encountered prior to the test? How were groups constructed during turns out? How familiar was each horse with the others?

The additional information on the familiarity between horse was given in lines 104-105.

LL120-124 You paired each horse with each. You have a total of 20 horses and the order of pairing (A vs B or B vs A) is not relevant. Therefore, you have a combinations issue (the way of selecting objects or numbers from a group of objects or collections, in such a way that the order of the objects does not matter – how many couples can we create having N horses?). To calculate it, the formula is nCr = n!/[r! (n-r)!], with nCr representing the number of combinations from “n” objects taken “r” at a time. So, in your case, 20!/2!(20-2)!=190 confrontations. You state you did 380 confrontations. What am I understanding incorrectly?

Thank you for that question. This is a mistake. There were 380 observations, and the number of confrontations is 190, you are perfectly right. It has been corrected in lines 118,162 and 184.

LL148-149 How did you assess that the professionals had “basic ethological knowledge”?

The raters were the teachers or undergraduate students of the Faculty of Animal Bioengineering, University of Warmia and Mazury in Olsztyn. One of the raters was a caretaker. This information was added in lines 142-148.

LL152-154 The questions included in the survey should be available for the reader.

The survey was added as Appendix 1.

L162 You state that you performed 379 tests, but the number was 380 in line 124.

Yes, it is still the same error, as above. Corrected.

Results

L184 You state that you performed 379 tests, but the number was 380 in line 124.

It is still the same error, as above. Corrected.

Table 3 P values are reported both as precise number and <0.01. I would choose one format only.

There were the values lower than 0.01. But we removed “<” as it is actually no difference in interpretation if there is 0.01 or lower.

In the caption, you reported that correlation at tendency level is “P<10”. Do you mean P<0.1?

Thank you, this is a mistake, we meant 0.10.

LL242-243 What “scientific sources” means in this context? Scientific papers? Textbooks? Others? I wouldn’t call “the trainers of ‘natural horsemanship’ or similar” behaviourists. They are trainers and their scientific competencies are diverse and often debatable.

 As the raters are scientific staff or students, they have academic knowledge of horse behaviour (added to the lines 142-148.). However, they declared that they gained the knowledge also from other sources, like behaviourists or NHT trainers. We completely agree that such so-called ‘behaviourists’ sell their philosophies, but we wanted to have an idea about the sources of knowledge of equines practitioners (this is not the subject of the present study). It is not known if they use this knowledge in practice, or if they just know the “behaviourist” philosophy. By the way, I (the last author) was scared of how effective the message sold to equine users and owners is. Even the persons with academic equine education accept it without any deeper consideration.

Discussion

I would like to see some discussion regarding practical applications of your findings. While instructors/riders may be aware of dominance ranks in their group of horses, during daily tasks there may be possible confounding factors, such as: number of horses during each class (there are no usually pairs, but multiple horses in the same arena), presence of people (do people influence horse behaviour? In which way? Do they influence their dominance/submission behaviour?), training (are horses trained to work with other horses, no matter the dominance ranks? Since the rider requests drive the horse behaviour, how does this influence the possible dominance related signals or behaviour?).

Thank you, these are very important questions. We liked them very much and we quoted them in the discussion. We put them at the end of the discussion as potential subjects of further studies, since we have no answers to them based on the results of our study.

References

Please check the references section for homogeneity.

The references were adjusted to the journal format.

Reviewer 2 Report

Comments and Suggestions for Authors

This is an interesting and unique study, with some useful observations. The writing is clear, but the manuscript could benefit from making some adjustments to the content – ie. ensuring each section contains the relevant information and relating the discussion more effectively to the aims of the study, which would then support your conclusions. Please ensure the methods are in the relevant section, and please explain what your results show – rather than just describing the statistics, please explain what they mean. More detailed comments are below:

Simple Summary

Line16 – spelling – should ‘practicians’ read ‘practitioners’ – please alter this throughout the document.

Line18-19. Unclear what the correlation of time and submissive behaviours have to do with rater ranking? This is explained in the abstract well, but perhaps needs rewording here to aid in clarity and context to the reader

Abstract

Introduction

Line 49 – do you mean that equine practitioners are expected to have a minimum knowledge? -consider rewording for clarity

Line 54 – do you mean that 2 year old horses leave the social group at 2 years old? Consider rewording to this effect, or clarifying the statement.

Line 65 &64 – why does the reference include e.g.? Please remove this throughout the document

Line 88 – instead of ‘contesting over an important…’ change to ‘contesting an important …’. Please change ‘persons’ to practitioners for continuity throughout the article,. Please provide a brief explanation of your hypothesis (after your aim) and why assessing the result will be important.

Lines 88-94 – These are methods. Please integrate these lines into the Methods section.

Methods

Line 100 – where does this figure come from (15 years) – either add a reference, or remove this to your results. Is this figure based on how long the horses in the study had been with the riding school, or is it historic data? If it is the study horses, please state this, and if reworded to explain this, it could remain in the methods.

Line 102 – please explain leisure riding (in the arena or hacking outside?)

Line 105 – please change ‘for’ to ‘in’

Lines 108-115 – This information should be placed either in the introduction as justification for your methods or maybe better in the discussion where you can defend your use of the feed confrontation test, and where the results may or may not accurately reflect reality. This would start better with the sentence “ In pairs…..”

Lines126: Was there any consideration to the time of day (feeding time, after/before work etc) that the feed confrontation test was conducted? Was it at a similar time of day for all tests? Some mention of this could be useful, and discussion about how that might affect your results (especially the amount of NI horses).

Line 132: Could you put your formulas in an equation for ease of reading?

Table 1: spelling – ‘pinnae’

Line 144-154: Were all 8 people working with the horses? – if one did not have ethological knowledge of horses, what was their role? I think this section might need a bit more explanation for clarity of how the raters were chosen, and what their experience was and how they were classified into groups.

Line 174 – was the difference between test and ignorant raters significantly different? (What is the p value between the groups?)

Table 2: please include units in the table where possible

Line 193: please explain rater F in more detail

Line 195: is -0.69 correct? Shouldn’t it be -0.65? Perhaps it would be more useful to give the mean or median values here, as you can see the range in the Table 3.

Table 3: is the p value for tendency level correct at 10? Were raters predicting incidences of behaviour? I’m not entirely sure what this table is portraying – could there be a bit of explanation of this in the methods  - ie. what variables the raters were asked to include – or their simple survey and questions relating to behaviour predictions could be added as supplementary information in the methods section.

Lines 201-214: This section is slightly confusing. I think a deeper explanation of the raters and their association with the horses (in the methods) might help this section – or, as it presents information already included in Table 3 – perhaps it would be better to shift this section (without repetition of the values) to the discussion with an explanation of why different raters might predict the different behaviours – or was this simply random ?

Discussion

The discussion presents some interesting points and interpretations. I think it would benefit from the addition of rater responses and how they differ between raters – as stated above. This would tie your discussion more effectively to your aims, which are only briefly addressed and tie into your conclusion. Please also include a limitations section.

Line 260-262 – this sentence is confusing, please consider rewording

Line 311: please remove ‘of’

Lines 327-340 – great!

References

A lot of these references are in books – could you find some relevant published studies to support your research?.

Please check your formatting of the references, eg. Reference 30 – journal name should be in italics.

Comments on the Quality of English Language

Very minor edits, please check grammar.

Author Response

Reviewer 2

We are grateful for your positive opinion. We hope the manuscript was improved.

This is an interesting and unique study, with some useful observations. The writing is clear, but the manuscript could benefit from making some adjustments to the content – ie. ensuring each section contains the relevant information and relating the discussion more effectively to the aims of the study, which would then support your conclusions. Please ensure the methods are in the relevant section, and please explain what your results show – rather than just describing the statistics, please explain what they mean. More detailed comments are below:

Simple Summary

Line16 – spelling – should ‘practicians’ read ‘practitioners’ – please alter this throughout the document.

Thank you, it has been corrected.

Line18-19. Unclear what the correlation of time and submissive behaviours have to do with rater ranking? This is explained in the abstract well, but perhaps needs rewording here to aid in clarity and context to the reader

These lines were reworded for more clarity (19-20).

Abstract

Introduction

Line 49 – do you mean that equine practitioners are expected to have a minimum knowledge? -consider rewording for clarity

We added that it is advisable they have such a knowledge (L50).

Line 54 – do you mean that 2 year old horses leave the social group at 2 years old? Consider rewording to this effect, or clarifying the statement.

It was corrected as suggested (L54)

Line 65 &64 – why does the reference include e.g.? Please remove this throughout the document

It was removed throughout the text.

Line 88 – instead of ‘contesting over an important…’ change to ‘contesting an important …’.

This was changed (L89).

Please change ‘persons’ to practitioners for continuity throughout the article,.

This was changed throughout the article.

Please provide a brief explanation of your hypothesis (after your aim) and why assessing the result will be important.

We added the a short explanation in lines 89-91.

Lines 88-94 – These are methods. Please integrate these lines into the Methods section.

These lines were deleted.

Methods

Line 100 – where does this figure come from (15 years) – either add a reference, or remove this to your results. Is this figure based on how long the horses in the study had been with the riding school, or is it historic data? If it is the study horses, please state this, and if reworded to explain this, it could remain in the methods.

We are not sure which figure you have in mind. Figure 1 represents the scheme of the test, the others are print screens of recordings. We added the “study” to the footnotes of figure 2 and 3.

Line 102 – please explain leisure riding (in the arena or hacking outside?)

We added that horses were mainly used for riding in the arena (L100-101)

Line 105 – please change ‘for’ to ‘in’

This was changed.

Lines 108-115 – This information should be placed either in the introduction as justification for your methods or maybe better in the discussion where you can defend your use of the feed confrontation test, and where the results may or may not accurately reflect reality. This would start better with the sentence “ In pairs…..”

This part was moved to discussion, lines 362-371

Lines126: Was there any consideration to the time of day (feeding time, after/before work etc) that the feed confrontation test was conducted? Was it at a similar time of day for all tests? Some mention of this could be useful, and discussion about how that might affect your results (especially the amount of NI horses).

The tests were conducted between 11 a.m. and 2 p.m., for 10 consecutive days, with a break for the weekend. This information was added to the methods (108-109).

Line 132: Could you put your formulas in an equation for ease of reading?

The formula was added (L128)

Table 1: spelling – ‘pinnae’

We are not sure about this comment: the spelling is the same as given by the reviewer.

Line 144-154: Were all 8 people working with the horses? – if one did not have ethological knowledge of horses, what was their role? I think this section might need a bit more explanation for clarity of how the raters were chosen, and what their experience was and how they were classified into groups.

Yes, all of them were working with horses or riding them (the students). More explanation was given in lines 149-154. The classification into the groups resulted from the fact that one teacher and three students participated in the experiment for the fulfilment of the MSc dissertation. The other persons were asked to rate the horses as they used them for riding lessons, and one was a groom.

Line 174 – was the difference between test and ignorant raters significantly different? (What is the p value between the groups?)

It is not clear about the difference in what? Do you mean the difference in dominance rank?  All raters used the whole spectrum of the rating scale.

Table 2: please include units in the table where possible

The unit (seconds and numbers) were added to the headline of table 2.

Line 193: please explain rater F in more detail

The results of the rater F are presented in lines 205, 212 and 215.

Line 195: is -0.69 correct? Shouldn’t it be -0.65? Perhaps it would be more useful to give the mean or median values here, as you can see the range in the Table 3.

Thank you, yes, this should be -0.65. We are not sure about the variable (-s) that has to be presented in mean or medians.

Table 3: is the p value for tendency level correct at 10?

This is an error, this should be P < 0.10 and it was corrected.

Were raters predicting incidences of behaviour? I’m not entirely sure what this table is portraying – could there be a bit of explanation of this in the methods  - ie. what variables the raters were asked to include

The raters were simply asked to rank the horses based on their experience/familiarity with them. No other instructions were given. Then, we correlated their rankings to see how it relates to the behaviours observed in the hay test to see if the rankings would be supported by observed social behaviours.

 – or their simple survey and questions relating to behaviour predictions could be added as supplementary information in the methods section.

The simple survey questions were added as Appendix 1. It was to simply evaluate their experience and the sources of their knowledge and this was not further examined. 

Lines 201-214: This section is slightly confusing. I think a deeper explanation of the raters and their association with the horses (in the methods) might help this section – or, as it presents information already included in Table 3 – perhaps it would be better to shift this section (without repetition of the values) to the discussion with an explanation of why different raters might predict the different behaviours – or was this simply random ?

In these lines, we just presented the results of how raters’ ratings predicted the behaviours of horses. More discussion on these results is in lines 274-280.

Discussion

The discussion presents some interesting points and interpretations. I think it would benefit from the addition of rater responses and how they differ between raters – as stated above.

The raters responses were presented in lines 235-247, and in the appendix1.

We also addressed the similarities/differences between raters as per dominance order and, consequently, related behaviours in lines 274-280.

This would tie your discussion more effectively to your aims, which are only briefly addressed and tie into your conclusion. Please also include a limitations section.

Line 260-262 – this sentence is confusing, please consider rewording

We removed the line about scavenger animals.

Line 311: please remove ‘of’

It has been corrected.

Lines 327-340 – great!

We are were glad you liked them.

References

A lot of these references are in books – could you find some relevant published studies to support your research?.

The references were adjusted when possible.

Please check your formatting of the references, eg. Reference 30 – journal name should be in italics.

The format of references was checked and adjusted.